# Temperature-Sensing Inks Using Electrohydrodynamic Inkjet Printing Technology

**DOI:** 10.3390/ma14195623

**Published:** 2021-09-27

**Authors:** Ju-Hun Ahn, Hee-Ju Hong, Chang-Yull Lee

**Affiliations:** 1Department of Aerospace Engineering, Inha University, Incheon 22212, Korea; jhahn@ssml.re.kr; 2Department of Aerospace Engineering, Chosun University, Gwangju 61452, Korea; hjhong@ssml.re.kr

**Keywords:** EHD inkjet printing, nanoceramic, silver nanoparticle, voltage, temperature sensor

## Abstract

Temperature measurement is very important for thermal control, which is required for the advancement of mechanical and electronic devices. However, current temperature sensors are limited by their inability to measure curved surfaces. To overcome this problem, several methods for printing flexible substrates were proposed. Among them, electrohydrodynamic (EHD) inkjet printing technology was adopted because it has the highest resolution. Since EHD inkjet printing technology is limited by the type of ink used, an ink with temperature-sensing properties was manufactured for use in this printer. To confirm the applicability of the prepared ink, its resistance characteristics were investigated, and the arrangement and characteristics of the particles were observed. Then, the ink was printed using the EHD inkjet approach. In addition, studies of the meniscus shapes and line widths of the printed results under various conditions confirmed the applicability of the ink to the EHD inkjet printing technology and the change in its resistance with temperature.

## 1. Introduction

Owing to recent developments in the field of artificial intelligence (AI), mechanical devices are increasing in complexity, along with the number of computations that electronic equipment must process. Although these developments have significantly advanced science and technology, they have also led to an increase in the amount of heat generated in mechanical devices and electronic equipment. This heat causes failure and fire, spurring research on devices that can be cooled rapidly [1,2,3,4]. Consequently, eco-friendly energy sources are attracting attention; in particular, extensive research on solar power generation is underway, despite the limited energy efficiency of solar heat [5]. Temperature sensors are commonly employed to regulate temperature. In general, temperature sensors are divided into non-contact and contact types. Although contact-type temperature sensors have high measurement accuracy, they are limited by their inability to measure curved surfaces because of their shape. Therefore, many temperature sensors must be arranged in a complicated manner to measure the temperature distribution. To overcome these shortcomings, a technique for manufacturing a sensor by printing it on a flexible substrate is proposed.

Printing technology, which is a patterning technique for fabricating structures on flexible substrates, has attracted significant attention owing to the recent growing interest in flexible electronic devices [6,7]. Current printing methods include inkjet printing, screen printing, and flexo printing. Inkjet printing, which is the most successful approach, has been extensively studied [8,9]. EHD inkjet printing technology falls under the inkjet printing category. Although the existing inkjet printing technology can achieve hundreds of microscopic results, EHD inkjet printing technology is considered a next-generation printing technology that is capable of achieving nanoscale results [10]. Yang et al. conducted a study to optimize polymethyl methacrylate (PMMA) nozzles [11]. Similarly, Cheng et al. conducted a study to minimize the clogging of PMMA nozzles through the capillary phenomenon [12]. Park et al. studied EHD inkjet printing with a high resolution [13]. Further studies to improve the development and function of inkjet printers are actively underway [14,15,16]. Studies were also conducted to produce ink that is suitable for EHD inkjet printing technology. Jang et al. fabricated a transparent electrode using an Ag grid [17]. Shabanov et al. studied copper inks, whose application to printing technology was difficult owing to oxidation, and produced copper inks that are capable of printing thin conductive copper wires [18]. As such, research on the application of EHD inkjet printing technology is actively underway [19,20].

Advanced printing technologies have been studied in various fields, such as sensors, transistors, and generators. Semiconductor devices produced using printing technology can be easily fabricated into the desired shapes and flexible devices. Furthermore, environment-friendly techniques can be used. Therefore, several technologies for manufacturing semiconductors using various printing technologies, including EHD inkjet printing technology, are being actively studied. Jung et al. researched the fabrication of an ion-gel transistor using the EHD process [21]. Sloma et al. printed carbon nanotubes and polyaniline and produced thermoelectric generators with flexible properties [22]. Wang et al. fabricated flexible proximity sensors using inkjet printing technology and succeeded in sensing temperature and vibrations [23]. A few studies have applied these advanced printing techniques to temperature-sensing paint. Most flexible sensors were mainly studied for measuring the parameters of vibration and pressure. However, research on temperature-sensing ink is very insufficient compared to research using piezoelectric sensor inks [24]. In the case of a temperature sensor, it can be applied to the skin, as well as in mechanical and electrical fields. Furthermore, it can be applied to the medical field.

In this work, inks with temperature-sensing capabilities were fabricated, and their properties were confirmed based on the ratio of nanoparticles they contained. The arrangement of particles was confirmed using scanning electron microscopy (SEM) imaging of the prepared ink, and variations in their resistance with the temperature were measured to confirm the characteristics of the ink. The ink with the best characteristics was employed in the EHD inkjet printer, and its applicability was confirmed experimentally.

## 2. Temperature-Sensing Ink Preparation and Measurement

### 2.1. Ink Preparation

Figure 1 shows the ink-preparation algorithm that was used for temperature sensing. First, diethylene glycol monobutyl ether acetate (DBGA; Daejung Chemicals & Metals, Seoul, Korea) was added to α-terpineol (Kanto Chemical, Tokyo, Japan), and the two solutions were mixed for approximately 1 h. Next, the nanoceramic particles in the form of powder were added to the mixed solution and dispersed for approximately 4 h to achieve temperature-responsive properties. Silver nanoparticles in the form of paste were then added and dispersed in the solution for approximately 20 h to provide electrical conductivity. The materials were added in different ratios to produce inks with varying resistance and printing characteristics to measure the temperature changes according to the mixing ratio. The solution that was obtained by mixing α-terpineol and DBGA, which was used as a solvent, was added in a certain amount. Based on the mixed solution, nanoceramics and silver nanoparticles were added in different proportions [25]. The dissolution ratio of the nanoceramics and silver nanoparticles was designed for nine conditions. These ratios are listed in Table 1.

### 2.2. Arrangement of Particles in the Ink

Figure 2 shows SEM images confirming the arrangement of the particles and the particle conditions. SEM imaging was performed at a magnification of 50,000× around the aggregated ceramic nanoparticles. Figure 2 shows the SEM images following the order listed in Table 1. The addition ratio of the silver nanoparticles increases from left to right, while the addition ratio of the nanoceramic particles increases from the top to the bottom. Further, the amount of added substances increases in the downward-right direction. Overall, particles with sizes between 0.4 and 1 μm were formed by the agglomeration of nanoceramic particles, and particles with a size of 0.1 μm or less were silver nanoparticles. The particles were arranged such that the silver nanoparticles were connected to the agglomerated ceramic particles. Therefore, as the amount of silver nanoparticles increased, the number of silver particles increased in the same area. As shown in Figure 2c, it is difficult to observe the agglomerated ceramic particles. A higher quantity of added material led to larger agglomerations of ceramic particles. The size of the agglomerated ceramic particles increased with the proportion of ceramic particles, similar to the increase in the amount of added material, and the number of nanoparticles increased in the same area.

### 2.3. Experimental Details

To confirm the characteristics of the prepared ink, the changes in the resistance were measured at a constant temperature. Figure 3 shows the schematic of the experimental setup for measuring the changes in resistance under various temperature conditions. To produce stable paints at high temperatures, inks were applied on glass plates to fabricate measurement substrates. Electrodes for facilitating resistance measurements were placed at both ends of the measurement substrate. The changes in resistance with temperature variations were measured in real time using the corresponding electrodes, and the values were collected through data acquisition (DAQ; Sefram, Saint-Étienne, France). During the experiment, the temperature in the chamber was increased continuously from 20 to 200 °C. The substrates for measurement were placed at the center of the chamber. The temperature around the substrate was measured by placing a thermocouple around it.

### 2.4. Ink Measurements

Figure 4 shows the change in the resistance as a function of the temperature. Figure 4 is arranged according to the order in Table 1. Among the inks manufactured in Figure 4g, the resistance was too large to exceed the range of the measuring devices. The difference between the lowest and highest resistances differed slightly for each ink. However, the positive temperature coefficient (PTC) property, where the resistance increases with temperature, was confirmed in the manufactured inks. Among the inks, the largest change in resistance occurred in Figure 4d,h, where the nanoceramic ratio was 1.5 times higher than that of the nanoparticles. In Figure 4c, where the ratio of silver nanoparticles was twice that of the nanoceramics, the resistance increased, after which, the PTC property was lost. Subsequently, the resistance decreased at approximately 160 °C. Therefore, the higher the ceramic proportion, the better the obtained ink; however, the addition of an excessive amount of ceramic caused a loss of electrical conductivity, leading to unsuccessful measurements.

## 3. EHD Inkjet Printing

### 3.1. Printing Details

Figure 5a shows a schematic of the EHD inkjet printing equipment. The printing equipment was controlled using a computer, and the motor system was controlled using the X, Y, and Z controllers. The motor systems affected the shape of the results and printing speed. The flow-rate controller controlled the amount of discharge while applying pressure to the syringe containing the inks. The high-voltage controller generated a high voltage between the nozzle and substrate and created an electric field that was used to generate different phenomena, as shown in Figure 5b. The appropriate behavior of each controller determined the shape of the meniscus formed at the tip of the nozzle. The discharge modes were divided into dripping, unstable cone jet, stable cone jet, and multijet, depending on the shape of the meniscus. The ideal discharge mode for EHD was the stable cone jet. The shape of the meniscus was examined using a charge-coupled device (CCD) camera.

### 3.2. Printability Confirmation

Printing was carried out to confirm the printing possibility. The printing speed was 30 mm/s and the flow rate was 3 μm/min. In addition, the experiment was conducted using a nozzle with an inner diameter of 0.15 mm. Figure 6 shows an enlarged photograph of the printing results of the previously prepared inks and a part of the printing results obtained using an optical microscope. The remaining inks were applied to the printer, except for ink (g), which was impossible to measure in the previous experiment. As a result of the printing, it can be seen that the patterning was formed in a straight line. The line width and particle density were determined using an optical microscope. The widths of the printed lines were in the range of 130–170 μm, and all the lines that were formed possessed very high particle densities. A high particle density resulted in increased conductivity, which is highly desirable.

### 3.3. Printing Results

Printing experiments were conducted using ink (d), which had the largest resistance change among the printable inks and showed a stable upward curve. Voltages ranging from 0 to 4 kV were applied to check the discharge stability. The printed line widths were compared when using flow rates from 2 to 5 µL/min. First, the discharge stability was evaluated. Figure 7 shows the meniscus shapes of the voltage for each flow rate. The meniscus in the low-voltage section shows the dripping shape under all conditions. This occurs when the effect of the surface tension of the inks was stronger than that of the voltage, and the inks fall in the shape of water droplets. Therefore, higher voltages were required. An unstable cone jet section appeared when the test was conducted by gradually increasing the voltage. Unstable cone jet sections occurred twice. In this state, the effect of the surface tension or voltage of the inks is slightly stronger, the meniscus has a cone jet shape, but the ejection direction is not constant. The stable cone jet is a section where the effects of the surface tension of the ink and the high voltage are balanced. In addition, it is the most ideal section for EHD inkjet printing technology. Its meniscus shape has a cone jet shape, and ejection occurs in a certain direction. Therefore, it is a section in which a thin wire can be formed by ejecting inks in the desired shape. Finally, the multijet is a section where electron movement in the ink occurs very actively because of the high voltage [26]. Therefore, it was confirmed that the stable cone jet, which is the most ideal ejection section, was most widely distributed at 4 µL/min. The stable cone jet section widened as the flow rate increased, with a maximum at 4 µL/min. Subsequently, this section was narrowed again.

Next, the line widths of the printing results according to the flow rate were compared. The voltage condition for printing was selected as 1.8 kV, which produced a stable cone jet with a meniscus shape under all flow conditions. As for the speed condition, the line width was printed relatively thickly, but printing was performed at 50 mm/s, which was a condition that produced stable line patterning. Figure 8 shows the printed line width as the flow rate increased. As the flow rate increased, the line width increased owing to the increased flow of ink. However, as the flow rate increased, the line width did not increase linearly. Printing results with a thinner line width could be obtained when the flow rate was lower, but the printing was observed to proceed very sensitively owing to the very small stable meniscus-shaped section.

## 4. Conclusions

To manufacture a temperature sensor using EHD inkjet printing technology, an experiment was conducted to produce the applicable ink. The manufacturing process focused on adjusting the experimental parameters according to the ratio of the materials used. When the temperature was increased, the manufactured ink had PTC properties. A higher ceramic ratio corresponded to a higher resistance change. However, when the proportion of nanoceramics was twice that of silver nanoparticles, the electrical conductivity was lost.

During the experiments, the ink with the best PTC characteristics was used for EHD inkjet printing. All experiments were conducted using the same printing conditions to confirm the applicability of the prepared inks. All inks, except those with the highest ratio of ceramic particles, were stably printed. In addition, using an optical microscope, the generation of high-density printing lines was confirmed. The EHD printing characteristics were confirmed using inks that showed the largest temperature change and were capable of stable printing. Among the many conditions of EHD inkjet printing technology, the meniscus characteristics were summarized using the voltage and flow rate, which had the greatest influence on the meniscus. In addition, the line widths were compared using the printing results in the section with a stable meniscus shape. The printing characteristics were confirmed through the results of two EHD inkjet printing experiments, and the applicability of the inks was confirmed.

## Figures and Tables

**Figure 1 materials-14-05623-f001:**
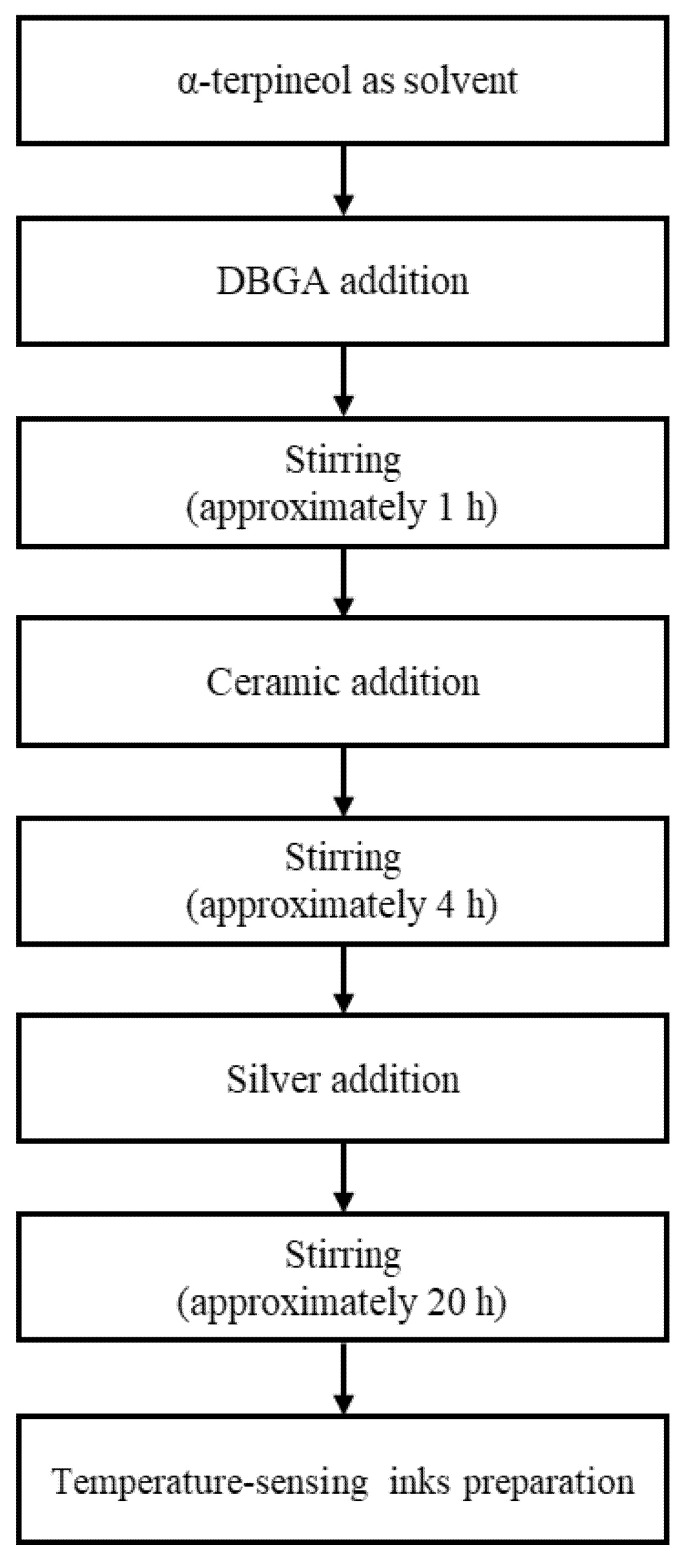
Flow chart of the manufacturing process for the temperature-sensing inks.

**Figure 2 materials-14-05623-f002:**
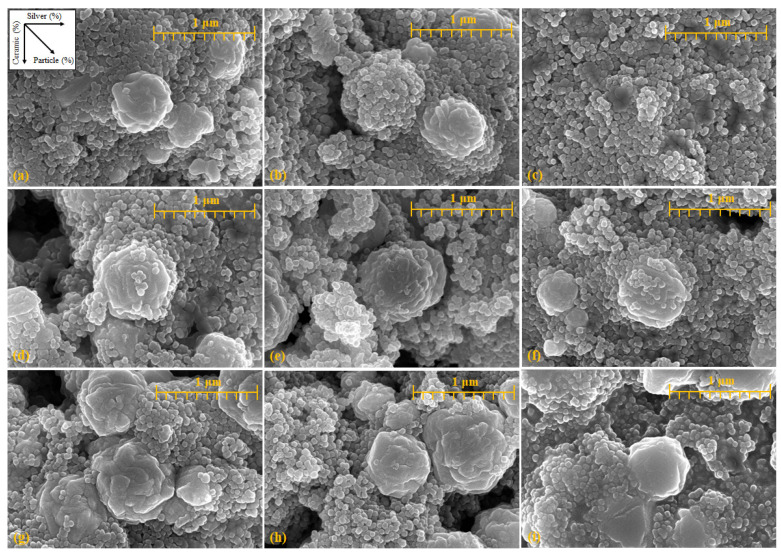
SEM images of various addition ratios for identifying particles (magnification: 50,000×): (**a**) 100:100, (**b**) 100:150, (**c**) 100:200, (**d**) 150:100, (**e**) 150:150, (**f**) 150:200, (**g**) 200:100, (**h**) 200:150, and (**i**) 200:200 (nanoceramics:silver nanoparticles).

**Figure 3 materials-14-05623-f003:**
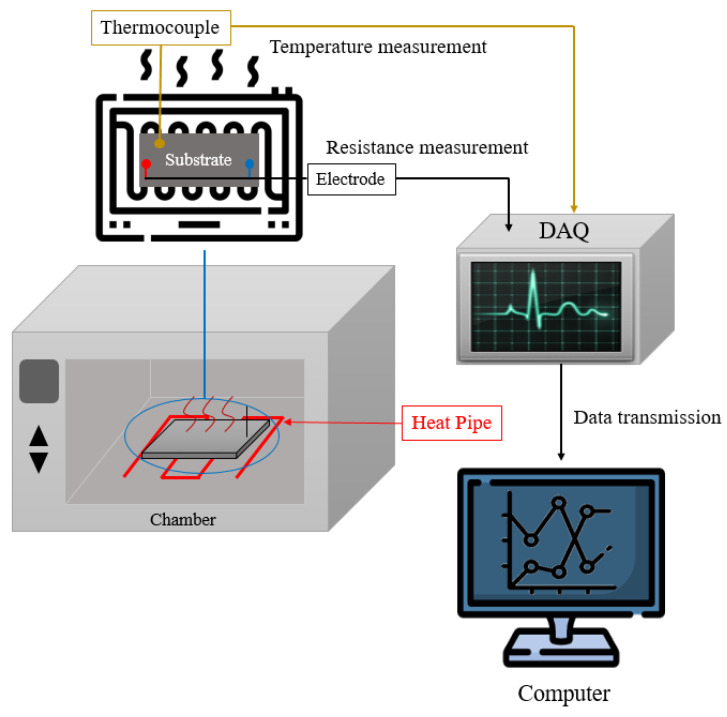
Schematic of the experimental setup for measuring the variation of the resistance with temperature.

**Figure 4 materials-14-05623-f004:**
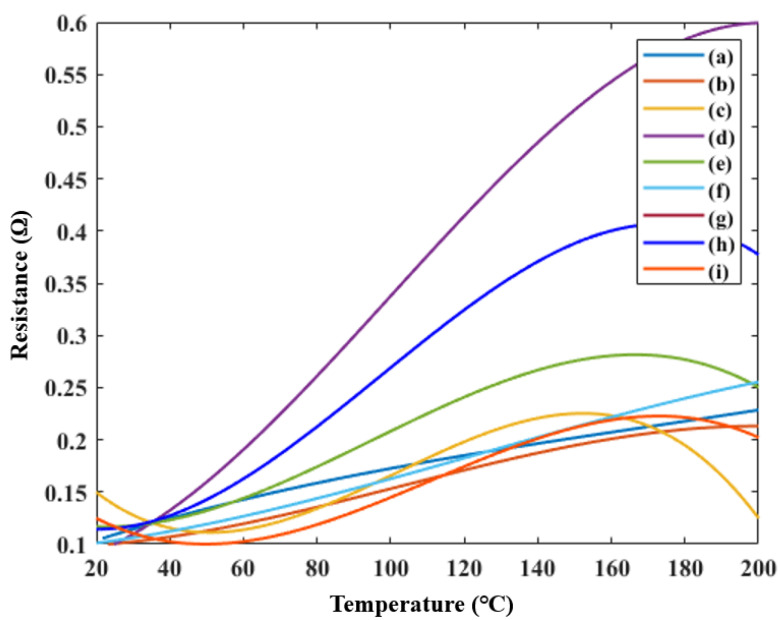
Resistances of the temperature-sensing inks with various material mixing ratios: (**a**) 100:100, (**b**) 100:150, (**c**) 100:200, (**d**) 150:100, (**e**) 150:150, (**f**) 150:200, (**g**) 200:100, (**h**) 200:150, and (**i**) 200:200 (nanoceramics:silver nanoparticles).

**Figure 5 materials-14-05623-f005:**
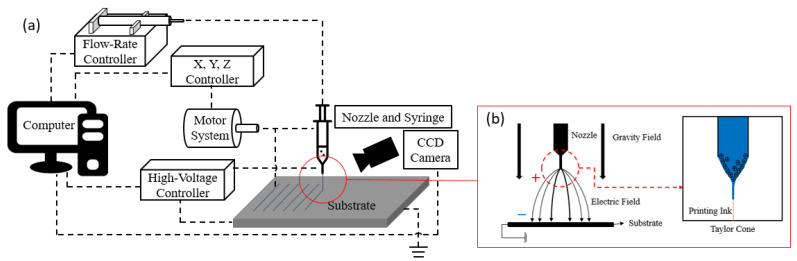
Schematic of the EHD inkjet printing system: (**a**) EHD Inkjet printing equipment, (**b**) Meniscus formation.

**Figure 6 materials-14-05623-f006:**
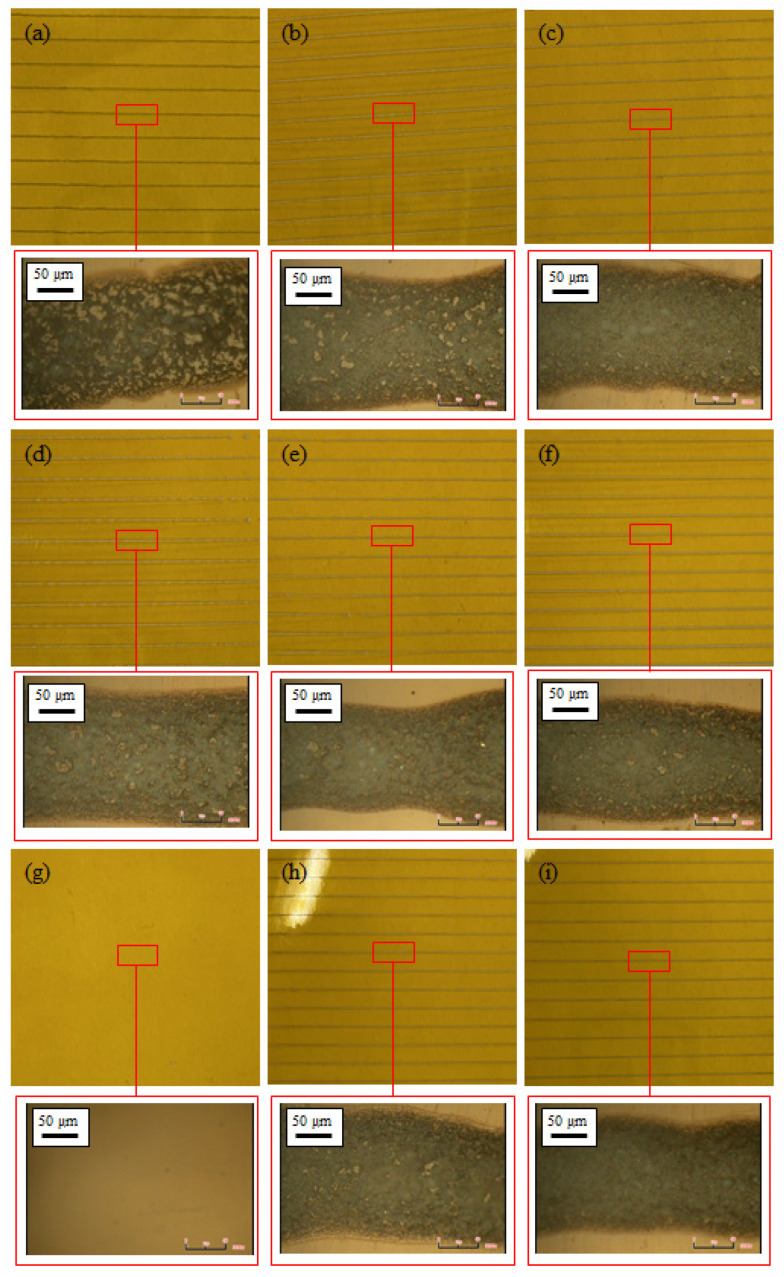
Ink printing results using the EHD inkjet printing and the particle density obtained using an optical microscope (magnification: 1200×): (**a**) 100:100, (**b**) 100:150, (**c**) 100:200, (**d**) 150:100, (**e**) 150:150, (**f**) 150:200, (**g**) 200:100, (**h**) 200:150, and (**i**) 200:200 (nanoceramics:silver nanoparticles).

**Figure 7 materials-14-05623-f007:**
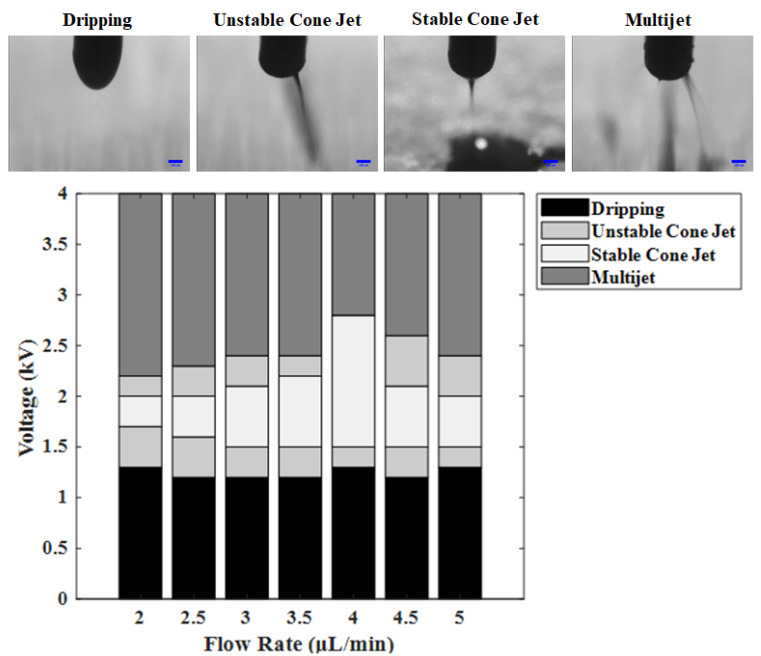
Meniscus stability diagram with flow rate–voltage condition and meniscus shape.

**Figure 8 materials-14-05623-f008:**
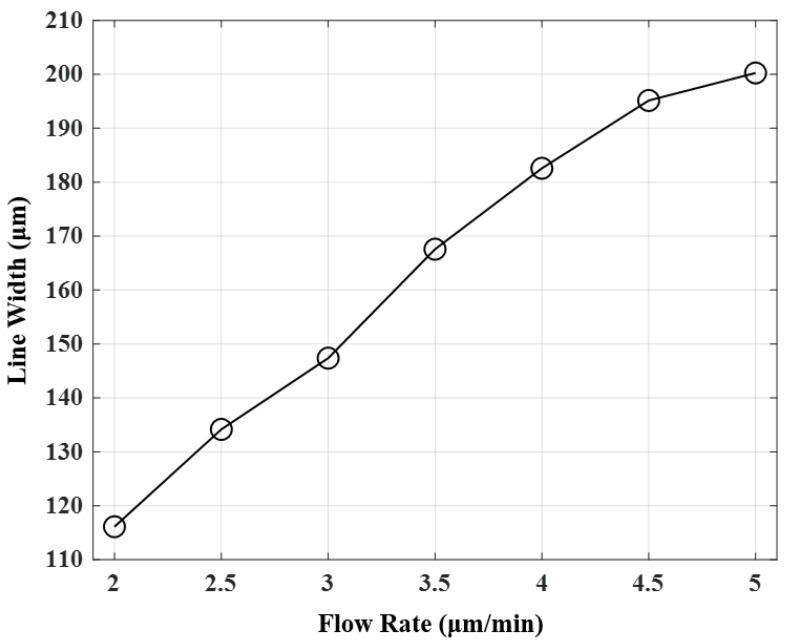
Line widths that were obtained for different flow rates during EHD inkjet printing.

**Table 1 materials-14-05623-t001:** Ratio of the nanoceramics and silver nanoparticles used in manufacturing the temperature-sensing inks.

Tag	α-terpineol + DBGA (%)	Ceramic (%)	Silver (%)
(a)	100	100	100
(b)	100	100	150
(c)	100	100	200
(d)	100	150	100
(e)	100	150	150
(f)	100	150	200
(g)	100	200	100
(h)	100	200	150
(i)	100	200	200

## Data Availability

The data are available upon request from the corresponding authors.

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
