# Peer review of "Temperature-Sensing Inks Using Electrohydrodynamic Inkjet Printing Technology"

_materials, 2021, doi:10.3390/ma14195623_

Round 1
Reviewer 1 Report
The manuscript entitled: “Temperature-sensing Inks Using Electrohydrodynamic Inkjet Printing Technology” authored by Ju-Hun Ahn and all, focused on formulation and characterization of the inks with temperature-sensing capabilities.
Below are some comments which in my opinion, could be useful.
Page 4 – Figure 2 – chemical composition to be better presented; add if possible Energy Dispersive X-Ray Analysis (EDX).
Page 5 line 131 the sentence “Among the prepared inks in Figure 4(g), 131 the ink with the highest nanoceramic ratio did not exhibit any electrical conductivity and thus could not be measured” is not clear; can authors present/explain this statement clearly? And what is the reason why this is happen.
Page 7, line 164, Printing was performed under arbitrary conditions to check for printability - i suggest the authors to add the deposition/printing details.
Page 8 line 186 the authors mentioned about the surface tension, it would be good if the inks ‘parameters such as viscosity, surface tension and pH will be incorporated in the paper. also the type of the nanoceramic material used in this work
Page 8, line 191/192 – the sentence “The stable cone jet is a section where ink and voltage are ejected with an appropriate force” is not clear can authors re-write it.
Page 8 line 194 “Finally, the multi-jet is a section where electron movement in the ink occurs very actively because of the high voltage” – can authors expand/add evidence or reference for this?
Reviewer 2 Report
1# Some experimental details should be provided to make sure others can repeat it. For example, the ratio of diethylene glycol monobutyl ether acetate (DBGA) to α-terpineol should be given. Are silver nanoparticles added in solid form or dispersion form? the diameter of silver nanoparticles should be provided. 2# In 2.4, the authors claimed that “Among the prepared inks in Figure 4(g), the ink with the highest nanoceramic ratio did not exhibit any electrical conductivity and thus could not be measured.” This description is not accurate. It should be “The resistance is too large to exceed the range of the measuring devices (or multimeter)” 3# In 3.1, the diameter of the nozzle used for printing process should be provided. 4# I think experiment should be step by step. The authors have found that ink (g) is high resistance in 2.4 Ink measurement, which is not suitable for printing process. I do not think it is necessary to do printing process with ink (g) in 3 EHD Inkjet Printing. 5# Scale bars should be provided in Figure 6. 6# Is there any applications provided using EHD inkjet printing temperature-sensing ink? I think proper applications should make “this story” more interesting. 7# Some papers should be cited to support the statement of the first sentence in the second paragraph of the 1. Introduction. For example, https://doi.org/10.1039/D1MA00463H and https://doi.org/10.1002/admt.201800546Author Response
Please see the attachment.

Round 2
Reviewer 2 Report
The revised manuscript can be accepted for publication.